# DisCo: Reinforcement with Diversity Constraints for Multi-Human Generation

## Abstract

State-of-the-art text-to-image models excel at realism but collapse on multi-human prompts—duplicating faces, merging identities, and miscounting individuals. We introduce DisCo (Reinforcement with DiverSity Constraints), the first RL-based framework to directly optimize identity diversity in multi-human generation. DisCo fine-tunes flow-matching models via Group-Relative Policy Optimization (GRPO) with a compositional reward that (i) penalizes intra-image facial similarity, (ii) discourages cross-sample identity repetition, (iii) enforces accurate person counts, and (iv) preserves visual fidelity through human preference scores. A single-stage curriculum stabilizes training as complexity scales, requiring no extra annotations. On the DiverseHumans Testset, DisCo achieves 98.6% Unique Face Accuracy and near-perfect Global Identity Spread—surpassing both open-source and proprietary methods (e.g., Gemini, GPT-Image) while maintaining competitive perceptual quality. Our results establish DisCo as a scalable, annotation-free solution that resolves the long-standing identity crisis in generative models and sets a new benchmark for multi-human image generation.

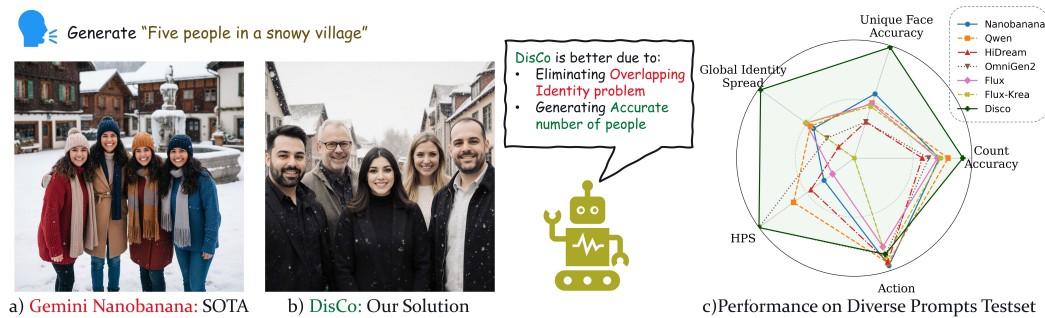

Figure 1: **DisCo enables identity-consistent multi-human generation.** (a) SOTA methods often produce duplicate or inconsistent faces, while (b) DisCo generates distinct, diverse identities. (c) Quantitative results show clear gains in Count Accuracy, Unique Face Accuracy, Identity Spread, and Overall quality(HPSv2 score).

## 1 Introduction

Text-to-image models have recently achieved impressive realism and controllability, powered by diffusion models (Ho et al., 2020; Rombach et al., 2022; Podell et al., 2024) and flow-based training schemes such as rectified flow and flow matching (Liu et al., 2022; Lipman et al., 2023). However, when tasked with generating *scenes with multiple people*, current systems frequently replicate nearly identical faces, conflate identities, or miscount individuals, undermining realism and limiting practical utility. This limitation was recently pointed out in Borse et al. (2025). This is a severe constraint in synthetic data generation for various applications such as training group photo personalization models, consistent character generation and storytelling, narrative media, educational content creation, and simulation environments for social interaction research. As illustrated in Fig. 2, these failures persist even when overall image quality is high, revealing a bottleneck in *identity differentiation* within and across generations. We term this fundamental issue as the *identity crisis*.

Existing text-to-image methods rely mainly on generating realistic and aesthetically pleasing humans (Labs & AI, 2025; Cai et al., 2025). These models do not address identity diversity—especially

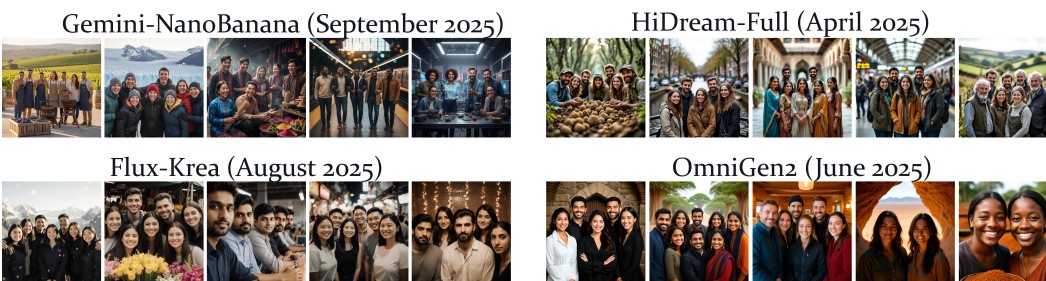

Figure 2: **The Identity Crisis.** Observe the images carefully, which have been generated by the recent SOTA text-to-image methods. From an initial glance, they look great. However, can you spot the issue?

as the number of people and scene complexity increase. We noticed that Reinforcement learning (RL) has been applied to the above models to optimize non-differentiable objectives such as prompt adherence, aesthetics, or human preferences (Black et al., 2023; Lee et al., 2023; Yang et al., 2024), and GRPO-style algorithms have improved stability and sample efficiency for flow-matching models (Liu et al., 2025; Xue et al., 2025). Additionally, RL has shown the ability to correct problematic behaviors that may be ingrained in large models through limited or biased training data—effectively breaking "bad habits" learned during pre-training. However, *no prior approach explicitly optimizes human-identity diversity both within a single image and across groups of generations for the same prompt.*

**We introduce DISCO—Reinforcement with DiverSity Constraints—a novel, sample-efficient RL framework for multi-human generation that directly targets identity diversity.** DISCO fine-tunes flow-matching text-to-image models using Group-Relative Policy Optimization (GRPO) (Liu et al., 2025; Xue et al., 2025), guided by a compositional reward that: (i) penalizes facial similarity within images, (ii) discourages repeated identities across groups, (iii) enforces count accuracy, and (iv) preserves text–image alignment via an HPS-style score. RL enables flexible optimization of heterogeneous, non-differentiable rewards, overcoming the limitations of supervised fine-tuning, which requires large, annotated datasets. To further enhance robustness as the number of people increases, DISCO employs a single-stage curriculum that anneals the prompt distribution from simpler cases to a uniform range (Liang et al., 2024).

**Empirically, DISCO sets a new standard for multi-human generation:** it substantially reduces identity duplication and improves fidelity across diverse prompts and model backbones (e.g., SDXL/SD3.5, FLUX variants, proprietary models), *without requiring auxiliary annotations*. On DiverseHumans and MultiHuman-TestBench, DISCO achieves consistent gains in *Count Accuracy* and *Unique-Faces/Non-overlapping Identity* while maintaining perceptual quality (Figs. 1, 5; Tables 1-2).

**Contributions.**

- **Identity and Count aware RL for multi-human scenes:** We cast multi-human generation as RL fine-tuning with diversity- and count-based rewards computed from facial embeddings, *within* images and *across* groups of generations.
- **Group-wise diversity reward:** We introduce a group-relative term that discourages cross-sample identity repetition, improving exploration and advantage estimation under GRPO.
- **Single-stage curriculum:** A lightweight sampling curriculum improves stability and generalization as the requested number of people scales.
- **State-of-the-art identity diversity with strong quality:** DISCO delivers large gains in identity uniqueness and count accuracy across models and prompts, *without extra spatial/semantic annotations*.

## 2 RELATED WORK

**Text-to-Image Generation.** Diffusion models (Ho et al., 2020) and latent diffusion (Rombach et al., 2022; Podell et al., 2024) have established high-fidelity text-to-image synthesis. Flow-

based formulations—rectified flow and flow matching—enable efficient, deterministic sampling with strong quality (Liu et al., 2022; Lipman et al., 2023; Labs, 2024; Labs & AI, 2025; Cai et al., 2025). Unified multimodal transformers integrate text and image tokens for subject-driven or reference-conditioned generation (Xiao et al., 2024; Xie et al., 2025; Mao et al., 2025; OpenAI, 2025; Wu et al., 2025). Despite these advances in realism and prompt alignment, *multi-human identity differentiation* remains a persistent failure mode in unconstrained scenes.

**Multi-Human Generation.** A NeurIPS 2025 study Borse et al. (2025) discuses the limitations the above methods on the multi-human generation task. They also identify the bias in Human generation by these models, also pointed out by Chauhan et al. (2024). In their future work section, they observed that current SOTA methods merge identities, repeat faces, or miscount people—the precise error modes DISCO targets (Fig. 2).

**Reinforcement Learning for Generative Image Models.** RL and preference-optimization have been used to optimize non-differentiable objectives such as prompt faithfulness, aesthetics, and human preferences (Black et al., 2023; Lee et al., 2023; Yang et al., 2024). In the flow-matching setting, GRPO provides value-free, group-relative variance reduction and KL-controlled updates, with curriculum and multi-objective extensions to improve stability and diversity (Liu et al., 2025; Xue et al., 2025). In contrast to prior work that largely optimizes faithfulness, **DISCO explicitly encodes facial-identity diversity constraints both intra-image and inter-image**, paired with an identity-aware curriculum, yielding robust gains in multi-human scenes while maintaining quality.

## 3 METHOD

In this Section, we discuss our proposed DISCO finetuning approach in detail. We begin by establishing the mathematical foundations in Section 3.1. Section 3.2 introduces our proposed compositional reward function. To handle the increasing complexity as the number of people generated grows, Section 3.3 presents a single-stage curriculum learning strategy that gradually transitions from simple to complex multi-person scenarios.

### 3.1 PRELIMINARIES

**Notation.** Let $c$ be a text prompt (conditioning), and $t \in [0, 1]$ index the sampling trajectory from noise ($t{=}1$) to data ($t{=}0$). The latent image distribution at time $t$ is denoted by $p_t(x)$, and the time grid by $\{t_k\}_{k=0}^{K}$ with $t_0{=}1 > \cdots > t_K{=}0$. We write $w_t$ for a standard $d$-dimensional Wiener process and use $\mathcal{N}(0, I)$ for the standard Gaussian.

**Flow matching and rectified flows.** We consider continuous-time normalizing flows trained with flow matching (FM) (Lipman et al., 2023). Given a data sample $x_0 \sim \mathcal{X}_0$ and noise $x_1 \sim \mathcal{N}(0, I)$, rectified flow (RF) Liu et al. (2022) defines the linear probability path

$$x_t = (1 - t)\, x_0 + t\, x_1, \quad t \in [0, 1], \tag{1}$$

and trains a velocity field $v_\theta(x_t, t)$ to regress the target velocity $v = x_1 - x_0$. FM yields efficient, deterministic ODE sampling with few steps and high sample quality.

**Denoising as an MDP.** We cast iterative sampling as an MDP $\langle \mathcal{S}, \mathcal{A}, \rho_0, P, R \rangle$ with state $s_k = (c, t_k, x_{t_k})$, action $a_k = x_{t_{k+1}}$, deterministic transition $s_{k+1} = (c, t_{k+1}, x_{t_{k+1}})$, and initial distribution $\rho_0(s_0) = (p(c), \delta_{t_0=1}, \mathcal{N}(0, I))$. The policy is $\pi_\theta(a_k \mid s_k) = p_\theta(x_{t_{k+1}} \mid x_{t_k}, c)$, and we compute a terminal reward $R(s_K) = r(x_{t_K}, c)$ at $t_K{=}0$ (e.g., Black et al., 2023; Yang et al., 2024).

**From ODE to Marginal-Preserving SDE.** We begin with the deterministic sampler defined by the probability-flow ODE:

$$\frac{dx_t}{dt} = v_\theta(x_t, t), \quad t \in [0, 1].$$

To enable exploration during RL while preserving marginals $\{p_t\}$, we follow Flow-GRPO Liu et al. (2025) and replace the ODE with an Itô SDE:

$$dx_t = f_\theta(x_t, t)\, dt + \sigma(t)\, dw_t, \tag{2}$$

Figure 3: **DISCO training overview.** Our method fine-tunes text-to-image models using Flow-GRPO with a compositional reward. Given a prompt, the model generates a group of images evaluated by four components: (1) *Intra-Image Diversity* penalizes duplicate identities within images, (2) *Group-wise Diversity* promotes variation across the group, (3) *Count Accuracy* enforces correct person count, and (4) *HPS Quality* ensures prompt alignment and visual fidelity. The combined reward guides GRPO updates to improve identity consistency and diversity.

which matches the same $p_t$ as the ODE. The relation between drift terms is:

$$v_\theta(x,t) = f_\theta(x,t) - \tfrac{1}{2}\sigma(t)^2 \nabla_x \log p_t(x),$$

allowing controlled stochasticity via $\sigma(t)$ and score-based compensation. We use Flow-GRPO's model-based score approximation; see Appendix D for details.

**Trajectory Policy and GRPO Objective.** Discretizing equation 2 over $K$ steps defines the trajectory policy $\pi_\theta(\tau \mid c) = \prod_k p_\theta(x_{t_{k+1}} \mid x_{t_k}, c)$, with log-probability $\log \pi_\theta(\tau \mid c) = \sum_k \log p_\theta(x_{t_{k+1}} \mid x_{t_k}, c)$. Returns $r(\tau, c)$ are computed on the final image $x_{t_K}$, with gradients back-propagated through all steps (Liu et al., 2025). For each prompt $c$, we sample a group $G = \{\tau_i\}_{i=1}^{M}$ and compute normalized advantages:

$$\tilde{A}_i = \frac{r(\tau_i, c) - \mu_c}{\sigma_c + \epsilon}, \quad \mu_c = \frac{1}{M}\sum_{i=1}^{M} r(\tau_i, c), \quad \sigma_c^2 = \frac{1}{M}\sum_{i=1}^{M}\big(r(\tau_i, c) - \mu_c\big)^2, \tag{3}$$

We optimize:

$$\max_\theta \ \mathbb{E}_c\left[\frac{1}{M}\sum_{i=1}^{M}\tilde{A}_i \log \pi_\theta(\tau_i \mid c)\right] \ - \ \beta_{KL}\,\mathbb{E}_c\big[\mathrm{KL}\big(\pi_\theta(\cdot \mid c) \,\|\, \pi_{\theta_{\mathrm{ref}}}(\cdot \mid c)\big)\big], \tag{4}$$

where $\pi_{\theta_{\mathrm{ref}}}$ is the frozen base model and $\beta_{KL}$ controls drift and reward hacking. For efficiency, we train with fewer denoising steps ($K_{\mathrm{train}} \ll K_{\mathrm{test}}$); full schedule is used at test time. See Appendix D for hyperparameters.

## 3.2 REWARD SIGNAL

Our goal is to train identity-aware generators that (i) avoid duplicate identities within an image, (ii) discourage reusing the same identity across samples of the same prompt, (iii) produce the requested person count, and (iv) preserve text-image quality/alignment. We therefore optimize a compositional reward evaluated at both image- and group-level. Given a prompt $c$ and a group $G = \{\tau_i\}_{i=1}^{M}$ of trajectories, the terminal image of trajectory $i$ is $x_i \equiv x_{i,t_K}$ and the total reward is

$$r(\tau_i, c, G) = \alpha\, r_{\mathrm{img}}^{d}(x_i) \ + \ \beta\, r_{\mathrm{grp}}^{d}(x_i, G) \ + \ \gamma\, r_{\mathrm{img}}^{c}(x_i) \ + \ \zeta\, r_{\mathrm{img}}^{q}(x_i), \tag{5}$$

with $\alpha, \beta, \gamma, \zeta > 0$. Unless stated otherwise, all four components are bounded in $[0, 1]$ to ensure a stable scale under GRPO. We detail each term below, highlighting robustness choices.

**Computing Facial Embeddings.** Each image $x_i$ is processed with RetinaFace Deng et al. (2019) Detector $D$, using a confidence threshold $\eta_{\mathrm{det}} = 0.7$, yielding bounding boxes $B_i = \{b_{i,j}\}_{j=1}^{m_i}$. Each face crop $\mathrm{crop}(x_i, b_{i,j})$ is encoded via ArcFace Deng et al. (2022) encoder $E$ to produce a $d$-dimensional embedding:

$$f_{i,j} = E\big(\mathrm{crop}(x_i, b_{i,j})\big) \in \mathbb{R}^d.$$

We denote the set of embeddings for image $i$ by $F_i = \{f_{i,1}, \ldots, f_{i,m_i}\}$. Identity similarity between embeddings $u, v \in \mathbb{R}^d$ is computed using cosine similarity $s(u, v) = \frac{u^\top v}{\|u\|_2 \|v\|_2}$, which simplifies to $u^\top v$ for $\ell_2$-normalized vectors. All similarity computations use $s(\cdot, \cdot)$ unless otherwise noted.

**Intra-Image Diversity** $r_{\text{img}}^d$. This component utilizes $\{F_i\}$ to enforce diversity by ensuring that the same individual does not appear multiple times within a single generated image.

$$r_{\text{img}}^d(x_i) = \begin{cases} 1 - \max_{j \neq k} s(f_{i,j}, f_{i,k}) & \text{if } m_i \geq 2 \\ 0.5 & \text{if } m_i < 2 \end{cases} \tag{6}$$

**Group-wise diversity** $r_{\text{grp}}^d$. Using this reward, we aim to discourage identity repetition across the group $G$ generated for the same prompt $c$. As the reward needs to be assigned per-image and not per-group, we compute the counterfactual "remove-one" statistic for every image $i$. Let $F_G = \bigcup_{i=1}^M F_i$ denote all faces across the group and define

$$S_G = \text{AvgPairwiseSim}(F_G) = \frac{2}{|F_G|(|F_G| - 1)} \sum_{\substack{i,j \in \{1,\ldots,|F_G|\} \\ i < j}} s(f_i, f_j) \in [0, 1]. $$

For image $i$, we remove its faces to get $F_{G-i}$ and compute $S_{G-i} = \text{AvgPairwiseSim}(F_{G-i})$. We define the contribution $\Delta_i = S_G - S_{G-i}$. If $S_{G-i} > S_G$ then $\Delta_i < 0$, meaning $i$ *increases* group diversity; we reward such samples. We map to $[0, 1]$ via

$$r_{\text{grp}}^d(x_i, G) = \sigma\big(-\lambda \Delta_i\big), \quad \sigma(u) = \frac{1}{1 + e^{-u}}, \quad \lambda = 5 \tag{7}$$

Pseudocode is provided in Appendix A.1. We observe the model performance generally increases when tuned with $r_{\text{img}}^d$ and $r_{\text{grp}}^d$. However, this model might be susceptible to **reward hacking**. The nature of hacking, illustrated in Appendix E.4, includes "grid" artifacts and generating lesser number of humans. Hence, we propose methods to regularize against them.

**Count Control** $r_{\text{img}}^c$. To ensure the appropriate number of distinct people and prevent generation of lesser faces, we use face count as a reward:

$$r_{\text{img}}^c(x_i) = \begin{cases} 1 & \text{if } m_i = N_{\text{target}} \\ 0 & \text{if } m_i \neq N_{\text{target}} \end{cases} \tag{8}$$

where $N_{\text{target}}$ is number of people in the prompt and $m_i$ is the number of faces detected.

**Quality/alignment term** $r_{\text{img}}^q$. To prevent the "grid" artifacts and facial distortions, we use HPSv3 Ma et al. (2025) as a reward. We normalize the HPSv3 score to $[0, 1]$:

$$r_{\text{img}}^q(x_i) = \tilde{q}(x_i) = \frac{\text{HPSv3}(x_i) - q_{\min}}{q_{\max} - q_{\min}}, \qquad q_{\min} = 0, \; q_{\max} = 10. \tag{9}$$

### 3.3 Single-stage Curriculum Learning

The difficulty of multi-human generation scales with the number of prompted faces. To handle this complexity, we apply curriculum learning that starts with simple scenarios (2-4 people) and gradually anneals to uniform sampling over the full range (2-$N_{\max}$ people). Let $\{\mathcal{P}_n\}_{n=2}^{N_{\max}}$ be prompts with $n$ people. Here, $N_{\max}$ is the max number of faces per prompt in training set. The sampling strategy at training step $t$ is:

$$p_t(n) = \begin{cases} p_{\text{annealed}}(n, t) & \text{if } t \leq t_{\text{curriculum}} \\ p_{\text{uniform}}(n) & \text{if } t > t_{\text{curriculum}} \end{cases} \tag{10}$$

where the annealing phase interpolates between simple and uniform distributions:

$$p_{\text{annealed}}(n, t) = \lambda_t \cdot p_{\text{uniform}}(n) + (1 - \lambda_t) \cdot p_{\text{simple}}(n), \tag{11}$$

$$p_{\text{simple}}(n) = \begin{cases} \frac{1}{3} & \text{if } n \in \{2, 3, 4\} \\ 0 & \text{otherwise} \end{cases}, \quad p_{\text{uniform}}(n) = \frac{1}{N_{\max} - 1} \tag{12}$$

with annealing weight $\lambda_t = \left(\frac{t}{t_{\text{curriculum}}}\right)^{\gamma_c}$, where $\gamma_c > 1$ controls how long the curriculum remains biased toward simple prompts. This strategy ensures gradual complexity increase from simple to uniform sampling across all prompt complexities. See A.2 for more details and D for hyperparams.

We apply DISCO finetuning to two models: a **generalist** (Flux-Dev) model and a **specialist** (Krea-Dev) model. Generalist models show lesser reliance on curriculum learning due to their broad training on diverse datasets. However, specialist models, optimized for specific aesthetics, benefit significantly from gradual complexity introduction. Curriculum learning is highly effective on the specialist model, as studied in Table 2.

### 3.4 DISCO ALGORITHM

We provide the complete Pseudocode for DisCo finetuning in Appendix A.3. For each update, we sample $n \sim p_t(\cdot)$, a prompt $c \in \mathcal{P}_n$, generate a group $G$ of $M$ trajectories under the SDE policy, detect faces and compute embeddings, evaluate rewards via Eqs. 6–9, compute advantages via equation 3, and update $\theta$ with equation 4. In the following Section, we discuss the Results of training using DisCo.

## 4 EXPERIMENT

### 4.1 EXPERIMENTAL SETUP

#### 4.1.1 DATASETS

**Training Data.** For training, we curated a dataset of 30,000 prompts containing group scenes with 2-7 people, with captions generated by GPT-5. The training prompts encompass diverse social contexts, settings, and activities including family gatherings, business meetings, recreational activities, and professional teams to ensure robust multi-human generation capabilities across varied scenarios.

**DiverseHumans.** For evaluation, we developed DiverseHumans, a comprehensive test set of 1,200 prompts systematically organized into six sections of 200 prompts each (corresponding to 2-7 people). Each prompt includes one of four diversity tag variants: no explicit diversity instruction (25%), general "diverse faces" instruction (25%), single ethnicity specification (25%), and individual ethnicity assignments for each person (25%). The dataset deliberately features different contexts from the training set to evaluate generalization capabilities, and for each prompt we generate multiple samples (typically 8-16) to assess both intra-image identity consistency and inter-image diversity.

**MultiHuman-TestBench.** We further evaluate on MultiHuman-TestBench (MHTB), an established recent benchmark introduced at NeurIPS 2025 for multi-human generation. MHTB provides comparison protocols on general multi-human generation capabilities without specific emphasis on identity diversity, and extend the scope of images to people performing simple and complex actions, complementing our DiverseHumans evaluation. Additional details are in Appendix B.

#### 4.1.2 MODELS

We compare against several baseline models including Nanobanana DeepMind (2025), SD3.5 AI et al. (2024), FLUX Labs (2024), Krea Labs & AI (2025), HiDream-Full Cai et al. (2025), Qwen-Image Wu et al. (2025), OmniGen2 Xiao et al. (2024), DreamO Mou et al. (2025) and GPT-Image OpenAI (2025). We fine-tune two open source models, FLUX-Dev(generalist) and Krea-Dev(specialist), using our DISCO framework to allow a direct performance comparison with their baseline counterparts. All implementation details and hyperparameters are provided in Appendix D.

#### 4.1.3 METRICS

To evaluate the performance of our model against the baseline, we report three key metrics: **Count Accuracy** measures the percentage of generated images that contain the exact number of individuals specified in the prompt. **Unique Face Accuracy (UFA)** quantifies the proportion of images in which all depicted individuals correspond to visually distinct identities, ensuring no duplicates within a

Table 1: Multi-Human Generation Evaluation. Results with * are possibly misleading, as the same MLLM is being probed to perform Generation and act as a judge. Green scores indicate the highest results and Red scores indicate the lowest results.

| Model | | Count Accuracy | Unique Face Accuracy (UFA) | Metrics Global Identity Spread (GIS) | HPS | Action Score | Average |
|---|---|---|---|---|---|---|---|
| **DiverseHumans-TestPrompts** (2-7 People) | | | | | | | |
| Proprietary | Gemini-Nanobanana | 72.3 | 57.2 | 42.7 | 31.9 | 95.7* | 60.0 |
| | GPT-Image-1 | 90.5 | 85.1 | 89.8 | 33.4 | 94.5 | 78.7 |
| Open-Source | HiDream | 57.9 | 32.3 | 16.2 | 32.2 | 92.4 | 46.2 |
| | Qwen-Image | 79.8 | 49.0 | 45.9 | 32.6 | 93.3 | 60.1 |
| | OmniGen2 | 63.3 | 32.3 | 28.7 | 33.4 | 86.2 | 48.8 |
| | DreamO | 70.5 | 31.8 | 35.2 | 32.0 | 82.7 | 50.4 |
| | SD3.5 | 55.3 | 69.1 | 72.5 | 28.1 | 71.3 | 59.3 |
| | Flux-Dev | 70.8 | 48.2 | 50.5 | 31.7 | 78.9 | 56.0 |
| | Krea-Dev | 73.6 | 45.8 | 50.6 | 31.2 | 87.9 | 57.8 |
| Ours | DISCO(Flux) | 92.4 | 98.6 | 98.3 | 33.4 | 85.6 | 81.7 |
| | DISCO(Krea) | 83.5 | 89.7 | 90.6 | 32.2 | 88.2 | 76.8 |
| **MultiHuman-TestBench** (1-5 People) | | | | | | | |
| Proprietary | Gemini-Nanobanana | 74.0 | 67.7 | 59.7 | 31.9 | 98.3* | 66.3 |
| | GPT-Image-1 | 90.7 | 83.7 | 81.0 | 33.2 | 96.2 | 77.0 |
| Open-Source | HiDream | 61.1 | 44.8 | 22.4 | 32.6 | 93.6 | 50.9 |
| | Qwen-Image | 80.3 | 47.9 | 50.6 | 33.2 | 94.5 | 61.3 |
| | OmniGen2 | 74.8 | 45.7 | 36.5 | 33.5 | 88.2 | 55.7 |
| | DreamO | 79.1 | 39.0 | 50.4 | 31.8 | 88.6 | 57.8 |
| | Flux-Dev | 61.8 | 56.5 | 51.2 | 31.4 | 88.5 | 57.9 |
| | Krea-Dev | 67.3 | 52.2 | 55.0 | 31.2 | 92.6 | 59.7 |
| Ours | DISCO(Flux) | 86.6 | 94.3 | 88.7 | 33.3 | 88.9 | 78.4 |
| | DISCO(Krea) | 83.8 | 80.1 | 84.1 | 32.9 | 92.3 | 74.6 |

single image. **Global Identity Spread (GIS)** is a global metric and assesses identity diversity across a dataset. by computing the ratio of total unique identities to the total prompted identities, in the testset. It indicates how effectively the model avoids repeating the same identities across different images. **HPSv2** assesses image quality and prompt/image alignment. We measure the MLLM **Action** scores for alignment with textual actions as proposed in MultiHuman-TestBench. See Appendix C for the full mathematical details.

## 4.2 RESULTS

### 4.2.1 QUANTITATIVE SCORES

**Diverse Humans Dataset.** Table 1 presents comprehensive evaluation results on the DiverseHumans-TestPrompts benchmark. Our DISCO approach demonstrates substantial improvements across all metrics compared to baseline models. DISCO(Flux) achieves 92.4% Count Accuracy versus baseline Flux's 70.8%, while DISCO(Krea) reaches 83.5% compared to Krea's 73.6%. The most significant gains are in UFA, where DISCO(Flux) reaches 98.6% versus 48.2% baseline, and DISCO(Krea) achieves 89.7% versus 45.8% baseline. Similarly, Global Identity Spread improves dramatically from 50.5% to 98.3% for Flux and from 50.6% to 90.6% for Krea. Notably, generalist models like Flux show larger absolute improvements than specialist models like Krea, though both benefit substantially from our approach. Remarkably, DISCO(Flux) surpasses even proprietary models like Nanobanana and GPT-Image-1 in Overall metrics, achieving superior UFA (98.6% vs 85.1%) and GIS (98.3% vs 89.8%).

Fig. 4 illustrates performance across varying numbers of individuals. While baseline models experience significant degradation as complexity increases, DISCO maintains consistently high performance. This robustness is particularly evident in UFA, where DISCO sustains above 90% accuracy even for scenes with 6-7 individuals, while baseline methods drop below 50%. This demonstrates DISCO's superior scalability. In panel (a), UFA performance shows DISCO does not produce overlapping identities even at high person counts, while baseline models exhibit a sharp drop. Panel

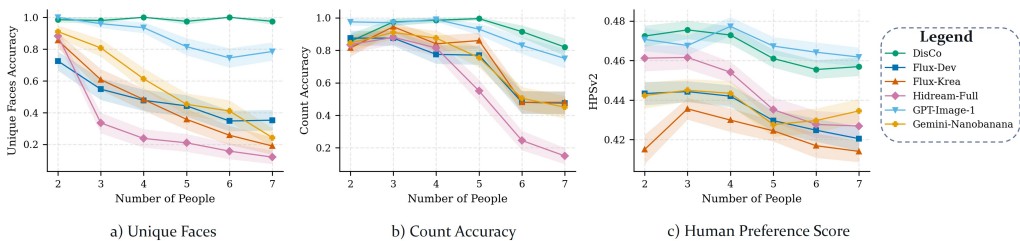

a) Unique Faces    b) Count Accuracy    c) Human Preference Score

Figure 4: **Performance vs. number of people.** We evaluate (a) Unique Face Accuracy, (b) Count Accuracy, and (c) HPSv2 across varying face counts. Error bars show 95% confidence intervals. DISCO(Flux)in  Green  consistently performs well across all metrics, maintaining high accuracy as face count increases.

(b) reveals similar trends for Count Accuracy. Panel (c) confirms that these improvements do not compromise perceptual quality, as HPS scores remain competitive across all configurations.

**MultiHuman-TestBench.** The MHTB results validate our findings across an independent dataset. DISCO(Flux) achieves 86.6% Count Accuracy and 94.3% UFA compared to baseline performance of 61.8% and 56.5% respectively, while DISCO(Krea) reaches 83.8% and 80.1% versus Krea's 67.3% and 52.2%. These consistent improvements across different evaluation protocols demonstrate the generalizability of our approach.

Importantly, over both datasets, HPS quality scores and MLLM Action scores show improvements over, or remain competitive with the respective (Flux/Krea) baselines. This demonstrates that our identity-focused optimization does not compromise overall generation quality or prompt adherence.

### 4.2.2 QUALITATIVE RESULTS

Fig. 5 showcases the clear visual improvements that DISCO brings to multi-human generation. Where baseline models struggle with repetitive faces and inaccurate person count, our approach delivers different individuals within each scene. Visualizing the examples, several patterns emerge that highlight DISCO's strengths. Most notably, we see an end to the identity crisis from Fig. 2, haunting SOTA methods. Instead, DISCO generates individuals with authentic variation in facial features, age, and appearance while preserving the natural demographic diversity we expect in real-world groups. The scenes maintain their coherence and visual appeal.

### 4.3 ABLATION STUDY

Table 2 ablates individual contributions of each DISCO component. This analysis is conducted on the Krea-Dev baseline, which proved more challenging to converge compared to Flux-Dev.

Table 2: Ablation Study: Progressive Addition of DISCO Components on Flux-Krea baseline

| Model | Rewards | | | | Curriculum | Metrics | | | |
| | Image Diversity | Group Diversity | Count Accuracy | HPS Score | | Count Accuracy | Unique Face Accuracy (UFA) | Global Identity Spread (GIS) | HPS Score |
|---|---|---|---|---|---|---|---|---|---|
| Krea | | | | | | 73.6 | 45.8 | 50.6 | 31.2 |
| +DisCo | ✓ | | | | | 66.2 | 78.6 | 50.8 | 31.7 |
| | ✓ | ✓ | | | | 67.3 | 80.2 | 72.5 | 32.0 |
| | ✓ | ✓ | ✓ | | | 81.1 | 83.2 | 68.3 | 31.9 |
| | ✓ | ✓ | ✓ | ✓ | | 79.2 | 82.6 | 73.7 | **32.4** |
| | ✓ | ✓ | ✓ | ✓ | ✓ | **83.5** | **89.7** | **90.6** | 32.2 |

Intra-image diversity dramatically improves unique face accuracy but leaves Global Identity Spread limited, as duplicate identities simply spread across different images rather than being eliminated. Adding group-wise diversity addresses this by enforcing diversity across the entire generation group, substantially improving cross-image identity variation.

Count accuracy drops when applying only group-wise rewards due to reward hacking—the model exploits generating fewer people as an easier optimization target. The count control component provides essential regularization, recovering count performance while maintaining identity diversity.

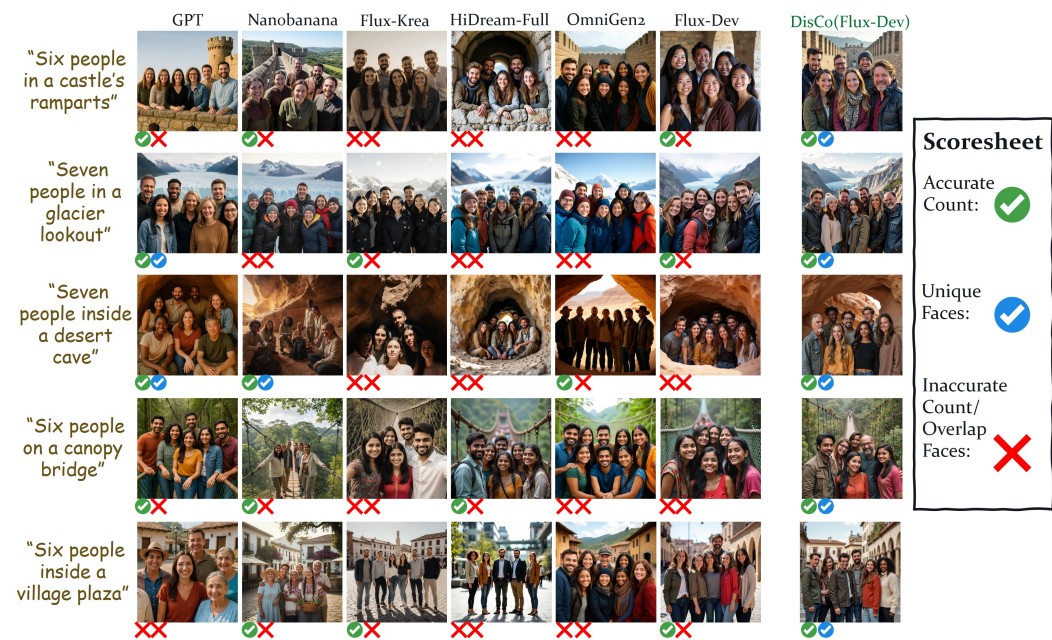

Figure 5: **DISCO vs. Related Work** DISCO finetuning improves performance over current SOTA methods to consistently generate accurate number of people without overlapping identity. It also maintains high perceptual quality while accurately following input prompts.

However, this introduces perceptual quality issues including unnatural "grid" arrangements of faces that technically satisfy requirements but appear artificial. HPS quality control effectively mitigates these artifacts by penalizing obvious visual anomalies.

The curriculum learning component delivers substantial improvements. Since Flux-Krea is not a generalist model, training convergence proved challenging without proper task decomposition. Curriculum learning addresses this by progressing from simple to complex scenarios, enabling the specialized model to learn the difficult multi-human generation task incrementally.

As evident from the scores, each component contributes meaningfully to the final performance, with the complete framework achieving optimal results across all metrics despite the challenging baseline characteristics.

## 5 CONCLUSION

Current state-of-the-art text-to-image models suffer from a fundamental *identity crisis* when generating multi-human scenes: they produce duplicate faces, conflate identities across individuals, and frequently miscount the requested number of people. We introduced DISCO, a reinforcement learning framework that directly targets this crisis through a novel compositional reward system that (i) penalizes intra-image facial similarity to eliminate duplicate identities, (ii) discourages cross-sample identity repetition to ensure diversity across generations, (iii) enforces accurate person counts, and (iv) preserves aesthetic quality and prompt alignment. By coupling GRPO fine-tuning with a principled single-stage curriculum, DISCO robustly solves the multi-human generation challenge while maintaining visual fidelity. Our empirical results demonstrate that DISCO not only resolves the identity crisis but achieves substantial performance improvements that surpass even proprietary models. On DiverseHumans, DISCO(Flux) achieves 98.6% Unique Face Accuracy—effectively eliminating identity duplication—compared to baseline Flux's 48.2% and proprietary Gemini-Nanobanana's 57.2%. Similar superiority holds across MultiHuman-TestBench, where DISCO(Flux) achieves 94.3% Unique Face Accuracy versus 56.5% baseline. Critically, these identity-focused optimizations enhance rather than compromise overall generation quality, establishing a new paradigm that pushes beyond existing proprietary model capabilities.

## ETHICS STATEMENT

Our work focuses on improving identity diversity in multi-human text-to-image generation to enhance fairness and realism in generative models. No human subjects, images or real identities were used; all experiments relied on (sanitized) text prompts and synthetic data. We anticipate positive societal benefits from our advancements in AI-driven multi-human image generation. By developing models that accurately generate diverse individuals across age, ethnicity, and gender, we aim to contribute to more equitable and inclusive digital media. Our work can enhance creative tools for artists and developers, enrich AR/VR/XR experiences, and improve assistive technologies. At the same time, we recognize potential risks, including misuse for misinformation campaigns or for impersonation. We also disclose the use of large language models (LLMs) for prompt generation, formatting assistance(for tables, plots), and text refinement. All generated outputs were carefully reviewed for quality and accuracy, and the scientific contributions, experiments, and conclusions remain the original work of the authors. We emphasize the importance of transparency, fairness audits, and responsible release practices, and strongly discourage malicious applications of this technology.

## REPRODUCIBILITY

To ensure reproducibility, we provide comprehensive implementation details as part of this submission. Our DISCO framework is implemented on top of the publicly available Flow-GRPO codebase, with training configurations specified in Appendix D (480 epochs, learning rate $1 \times 10^{-4}$, compositional reward weights ($\alpha = 0.50, \beta = 0.10, \gamma = 0.15, \zeta = 0.15$), and curriculum parameters ($\gamma = 2.0, t_{\text{curriculum}} = 40{,}000$ steps)). Appendix A provides complete algorithmic descriptions and pseudocode for group-wise diversity computation (Algorithm 1), curriculum learning (Algorithm 2), and the full DISCO training procedure (Algorithm 3). We also reference the publicly available detector and face embedding models. Our training dataset and the DiverseHumans evaluation set of 1,200 prompts are described in Appendix B, along with the (publicly available) MultiHuman-TestBench dataset used for evaluation. All evaluation metrics (Count Accuracy, Unique Face Accuracy, Global Identity Spread) are mathematically defined in Appendix C, with explicit similarity thresholds ($\kappa_{\text{dup}} = 0.5$) and clustering procedures. Baseline model evaluations follow official hyperparameters as documented in Appendix D, ensuring fair comparison. Finally, our distributed training setup (21 H100 GPUs with specified batch sizes and gradient accumulation) is fully documented in Appendix D to facilitate replication of our results.

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
