# Appendices

## APPENDIX CONTENTS

## A  EXTENDED METHOD

The following algorithms provide detailed pseudocode implementations of the key components described in Section 3. Algorithm 1 formalizes the group-wise diversity computation from Section 3.2, Algorithm 2 details the curriculum learning strategy from Section 3.3, and Algorithm 3 presents the complete training procedure that integrates all components from Section 3.4.

### A.1  GROUP-WISE DIVERSITY ALGORITHM

---
**Algorithm 1** Group-Level Identity Diversity Computation

---
**Require:** Group $G = \{x_i\}_{i=1}^{M}$, face embeddings $\{F_i\}_{i=1}^{M}$, scaling parameter $\lambda$

    $F_G \leftarrow \bigcup_{i=1}^{M} F_i$ {All faces across group}

    $S_G \leftarrow \text{AvgPairwiseSim}(F_G)$ {Baseline group similarity}

    **for** $i = 1$ to $M$ **do**

        $F_{G-i} \leftarrow F_G \setminus F_i$ {Remove faces from image $i$}

        $S_{G-i} \leftarrow \text{AvgPairwiseSim}(F_{G-i})$ {Similarity without image $i$}

        $\Delta_i \leftarrow S_G - S_{G-i}$ {Image $i$'s contribution to similarity}

        $r_{\text{grp}}^{d}(x_i, G) \leftarrow \sigma(-\lambda \cdot \Delta_i)$ {Sigmoid mapping with $\sigma(u) = \frac{1}{1+e^{-u}}$}

    **end for**

    **return** $\{r_{\text{grp}}^{d}(x_1, G), \ldots, r_{\text{grp}}^{d}(x_M, G)\}$

---

Algorithm 1 provides the implementation details for the counterfactual reward computation described in Section 3.2. The algorithm efficiently computes the baseline similarity $S_G$ once per group, then performs $M$ leave-one-out evaluations to determine each image's diversity contribution $\Delta_i$. In practice, with typical group sizes of $M = 21$ and face counts of 2-7 per image, the algorithm executes efficiently within the GRPO training loop.

### A.2  SINGLE-STAGE CURRICULUM LEARNING ALGORITHM

---

**Algorithm 2** DISCO: Single-stage Curriculum Learning

---

**Require:** Prompt sets $\{\mathcal{P}_n\}_{n=2}^{N_{\max}}$, curriculum parameters $t_{\text{curriculum}}$, $\gamma_c$

  Initialize training step $t = 0$

  **while** training not converged **do**

    **if** $t \leq t_{\text{curriculum}}$ **then**

      $\lambda_t \leftarrow \left( \frac{t}{t_{\text{curriculum}}} \right)^{\gamma_c}$ {Exponential annealing weight}

      **for** $n = 2$ to $N_{\max}$ **do**

        **if** $n \in \{2, 3, 4\}$ **then**

          $p_{\text{simple}}(n) \leftarrow \frac{1}{3}$

        **else**

          $p_{\text{simple}}(n) \leftarrow 0$

        **end if**

        $p_{\text{uniform}}(n) \leftarrow \frac{1}{N_{\max}-1}$

        $p_t(n) \leftarrow \lambda_t \cdot p_{\text{uniform}}(n) + (1 - \lambda_t) \cdot p_{\text{simple}}(n)$

      **end for**

    **else**

      **for** $n = 2$ to $N_{\max}$ **do**

        $p_t(n) \leftarrow \frac{1}{N_{\max}-1}$ {Uniform sampling}

      **end for**

    **end if**

    Sample $n \sim p_t(\cdot)$

    Sample prompt $c$ from $\mathcal{P}_n$

    Generate group $G$ and update model with prompt $c$

    $t \leftarrow t + 1$

  **end while**

---

Algorithm 2 provides the implementation details for the exponential curriculum strategy outlined in Section 3.3. The gamma parameter $\gamma_c$ controls the steepness of complexity introduction, with higher values maintaining focus on simple prompts for longer durations before transitioning to the full complexity range. The curriculum duration $t_{\text{curriculum}}$ determines the absolute training steps allocated to gradual complexity introduction before switching to uniform sampling across all prompt types. We define scenarios with 2-4 people as "simple" based on empirical analysis of baseline model performance degradation patterns. As shown in Figure 4, both Count Accuracy and Unique Face Accuracy exhibit the most pronounced performance drops at the 4-person threshold, with steeper degradation beyond this point, motivating our curriculum design that focuses initial training on these manageable scenarios before introducing the full complexity range.

### A.3 DISCO ALGORITHM

---

**Algorithm 3** DISCO: Overall Algorithm

---

**Require:** Pretrained flow-matching model $v_{\theta_0}$, prompt dataset $\mathcal{P}$, curriculum parameters $\eta$, $t_{\text{start}}$, $t_{\text{end}}$, reward weights $\alpha, \beta, \gamma, \zeta$

  **while** not converged **do**

    Sample $n \sim p_t(\cdot)$ and prompt $c \in \mathcal{P}_n$ using Algorithm 2

    Generate group $G = \{\tau_i\}_{i=1}^M$ using SDE policy $\pi_\theta(\cdot|c)$

    Extract facial embeddings: $F_i = \{E(\text{crop}(x_i, b)) : b \in D(x_i)\}$ for all $i$

    Compute compositional rewards: $r(\tau_i, G) = \alpha r_{\text{img}}^d + \beta r_{\text{grp}}^d + \gamma r_{\text{img}}^c + \zeta r_{\text{img}}^q$

    Compute group-normalized advantages $\{\tilde{A}_i\}$ and update $\theta$ using GRPO objective

    $t \leftarrow t + 1$

  **end while**

  **return** Fine-tuned model $\theta$

---

Algorithm 3 integrates all components described in Section 3 into the complete DISCO training procedure. The reward weights $\alpha, \beta, \gamma, \zeta$ control the relative importance of intra-image diversity, group diversity, count accuracy, and quality objectives respectively, allowing fine-grained control

over the optimization priorities. The group size $M$ determines the number of trajectories generated per prompt, directly affecting both the quality of group-normalized advantage estimation and the computational cost per training iteration.

# B DATASET DETAILS

## B.1 TRAINING DATASET

Our training dataset consists of 30,000 carefully curated prompts designed to capture diverse multi-human scenarios. Each prompt describes group scenes containing 2-7 people engaged in various activities and contexts. The captions were generated using GPT-5 to ensure high-quality, diverse descriptions that encompass a wide range of:

- **Social contexts**: Family gatherings, business meetings, friend groups, professional teams, recreational activities
- **Settings**: Indoor and outdoor environments, formal and informal occasions, workplace and leisure contexts
- **Activities**: Collaborative tasks, social interactions, professional activities, recreational pursuits
- **Group compositions**: Varying numbers of individuals (2-7) with diverse demographic representations

The prompts were designed to avoid overlap with evaluation datasets while maintaining sufficient diversity to train robust multi-human generation capabilities. The following are 5 examples of these prompts.

- Seven people on the desert dunes, hazy sun, diverse faces, clear faces visible, studio-quality, vivid detail
- Six people in an astronomy studio, Clean composition, Professional portrait, Portrait photography, Soft shadows, Natural lighting, Even exposure
- Three people in an aviation observatory, Sharp focus, Clean composition, Bokeh background, Color graded, Smiling expressions, Well lit
- Five people in a dawn-lit bakeshop, Studio quality, Even exposure, Group harmony, Cinematic lighting, Portrait photography, Soft shadows
- Seven people on a coastal boardwalk, afternoon light, diverse faces, clear faces visible, ultra-realistic, 8K resolution

## B.2 EVALUATION DATASETS

### B.2.1 DIVERSEHUMANS TEST SET

We developed DiverseHumans, a comprehensive evaluation dataset of 1,200 prompts specifically designed to assess identity consistency and diversity in multi-human generation. The dataset is systematically organized as follows:

**Structure**: Six sections of 200 prompts each, corresponding to scenes with 2, 3, 4, 5, 6, and 7 people respectively.

**Diversity Tags**: Each prompt includes one of four diversity specification levels:

1. **No tag** (25% of prompts): Basic scene descriptions without explicit diversity instructions
2. **"Diverse faces" tag** (25% of prompts): General diversity encouragement
3. **Single ethnicity specification** (25% of prompts): Mentions one of six racial/ethnic categories
4. **Individual ethnicity assignments** (25% of prompts): Specific ethnicity assigned to each person

**Example Prompts**:

- *No tag*: Five people on a island cove beach, High dynamic range, Group harmony, Professional portrait, Natural lighting, Smiling expressions
- *Diverse faces*: Five people in a antique arcade, High dynamic range, Sharp focus, Group harmony, Clear faces, Smiling expressions, Diverse faces among people
- *Single ethnicity*: Five people in a sidewalk cafe, Sharp focus, Bokeh background, Well lit, Clear faces, Group harmony, Indian ethnicity
- *Individual assignments*: Five people in a coastal market, Bokeh background, High dynamic range, Sharp focus, Professional portrait, Portrait photography, One person is White, One person is Middle-eastern, One person is Asian, One person is Black, One person is Hispanic

**Context Differentiation**: The DiverseHumans prompts deliberately feature different contexts and scenarios compared to the training set to evaluate generalization capabilities and prevent overfitting to training distributions.

### B.2.2 MULTIHUMAN-TESTBENCH (MHTB)

We additionally evaluate on the established MultiHuman-TestBench, a standardized benchmark for multi-human generation that provides consistent evaluation protocols and enables fair comparison with existing methods. MHTB focuses on general multi-human generation capabilities without specific emphasis on identity diversity, complementing our DiverseHumans evaluation. MHTB also asks for people performing specific actions (cooking, boxing, dancing, etc.) ranging from simple to complex, which is a key differentiator to DiverseHumans testset. We use their official implementation[1] to download data and compute metrics.

## C EVALUATION METRICS

To comprehensively evaluate multi-human generation performance as described in Section 4, we employ three core metrics that capture different aspects of identity consistency and counting accuracy. All metrics are computed using facial embeddings extracted via RetinaFace detection followed by ArcFace encoding, as detailed in our reward computation pipeline. All metrics are reported as percentages.

**Count Accuracy.** This metric measures the percentage of generated images that contain the exact number of individuals specified in the input prompt. For a given prompt $c$ with target count $N_{\text{target}}(c)$ and evaluation set $\mathcal{X}$, Count Accuracy is defined as:

$$\text{Count Accuracy (\%)} = 100 \times \frac{1}{|\mathcal{X}|} \sum_{x \in \mathcal{X}} \mathbf{1}\{F(x) = N_{\text{target}}(c)\}$$

where $F(x) = |D(x)|$ represents the number of detected faces in image $x$ using RetinaFace with confidence threshold $\kappa_{\text{det}} = 0.7$.

**Unique Face Accuracy (UFA).** This metric quantifies the percentage of images in which all depicted individuals correspond to visually distinct identities, ensuring no duplicate faces within a single image. We define faces as duplicates if their cosine similarity exceeds a threshold. Specifically, within image $x$, duplicates exist if:

$$\exists i \neq j : s(f_i, f_j) \geq \kappa_{\text{dup}}$$

where $s(\cdot, \cdot)$ denotes cosine similarity between face embeddings. The UFA metric is then computed as:

$$\text{UFA (\%)} = 100 \times \frac{1}{|\mathcal{X}|} \sum_{x \in \mathcal{X}} \mathbf{1}\{\text{no duplicates in } x\}$$

We set $\kappa_{\text{dup}} = 0.5$.

---

[1] https://github.com/Qualcomm-AI-research/MultiHuman-Testbench

**Global Identity Spread (GIS).**   This metric assesses identity diversity across an entire dataset of generated images by measuring the percentage of unique identities created relative to the total number of people requested across all prompts. For a batch $\mathcal{X}$ of images generated from prompts with respective target counts $\{N_{\text{target}}(c_i)\}$, we first cluster all face embeddings $\bigcup_{x \in \mathcal{X}} F(x)$ using single-linkage clustering with threshold $\kappa_{\text{dup}} = 0.5$. Let $C$ denote the total number of unique clusters (identities) found. The Global Identity Spread is then computed as:

$$\text{GIS (\%)} = 100 \times \frac{C}{\sum_i N_{\text{target}}(c_i)}$$

where the denominator represents the total number of people requested across all prompts in the batch. Higher GIS values indicate better identity diversity, with perfect diversity yielding GIS = 100% when every requested person has a unique identity.

**Action Score.**   We use the Action score as implemented in the MultiHuman-TestBench Borse et al. (2025) paper. This is an MLLM metric, which prompts Gemini-2.0-Flash using the image, and asks if the people in the image are performing the Action requested by the prompt.

**HPSv2:**   Due to our use of HPSv3 as a reward, we use the HPSv2 model to measure perceptual quality and prompt alignment. This step is to make the comparison with other methods fair, which may or may not have been trained with an HPSv3 reward.

# D   IMPLEMENTATION DETAILS

**DISCO Training.**   We implement DISCO using the public `flow_grpo`[2] framework with Flux pipeline, training in bf16 mixed precision on 512×512 images. Training uses 14 timesteps for reward computation and 28 steps for evaluation, with classifier-free guidance of 4.5 for Flux-Krea and 3.5 for Flux-Dev. We train for 480 epochs with batch sizes of 3 (train) and 16 (test), with a group size of 21. The compositional reward function combines intra-image diversity ($\alpha = 0.50$), group-wise diversity ($\beta = 0.10$), count accuracy ($\gamma = 0.15$), and HPS quality ($\zeta = 0.15$) components, with KL regularization weight $\beta_{KL} = 0.01$ to stabilize learning. We apply the proposed curriculum with $t_{\text{curriculum}} = 60$ epochs, and $\gamma_c = 3$

Training is distributed across 21 GPUs on 3 H100 clusters, with a single dedicated GPU for HPSv3 reward (3 nodes, 7 GPUs per node for training, 1 GPU as the HPSv3 server). We use a learning rate of $1 \times 10^{-4}$ with EMA enabled and checkpoint every 30 epochs. The curriculum learning strategy transitions from simple to complex prompts using exponential weighting parameter $\eta = 2.0$, with transition period from steps 10,000 to 40,000. Face detection uses RetinaFace (Deng et al., 2019) with confidence threshold 0.7, followed by ArcFace (Deng et al., 2022) embeddings for identity similarity computation. Total training time to 480 epochs is **13 hours**.

**Baseline Model Evaluation Settings.**   For fair comparison, we evaluate all baseline models using their recommended hyperparameters from official documentation. For OmniGen2, we use 50 inference steps with text guidance scale of 2.5 and image guidance scale of 3.0 for multi-modal tasks.[3] We set FLUX-Dev to 50 timesteps with CFG guidance of 3.5, while for FLUX-Krea we use 28 timesteps with CFG guidance 4.5 as specified in the official repository.[4][5] For SD3.5-Large, we apply 40 timesteps with guidance scale of 4.5.[6] We configure HiDream-I1 Full model with 50 timesteps and guidance scale 5.0.[7] We use 12 timesteps for DreamO and CFG guidance 4.5.[8]. We generate all images at 1024×1024 resolution. We set a different seed for every image (the image index itself), and we share these seeds across all evaluations.

---

[2]https://github.com/yifan123/flow_grpo
[3]https://huggingface.co/OmniGen2/OmniGen2
[4]https://huggingface.co/black-forest-labs/FLUX.1-dev
[5]https://github.com/krea-ai/flux-krea
[6]https://huggingface.co/stabilityai/stable-diffusion-3.5-large
[7]https://huggingface.co/HiDream-ai/HiDream-I1-Full
[8]https://github.com/bytedance/DreamO

# E    EXTENDED RESULTS

## E.1    QUANTITATIVE RESULTS

The quantitative results presented in this section provide detailed analysis of DISCO's performance across various experimental conditions and model configurations. These results complement the main paper findings by examining performance variations across different prompt types, reward weight configurations, and computational efficiency metrics.

### E.1.1    PERFORMANCE ON VARIOUS DIVERSITY TAGS IN PROMPTS

Table E.1 analyzes performance across the four diversity specification levels in our DiverseHumans dataset. The results reveal interesting patterns that demonstrate DISCO's effectiveness in addressing different types of diversity challenges.

For Unique Face Accuracy, baseline models show variable performance across diversity tags, with some models (like Gemini-Nanobanana) performing significantly better on explicit diversity prompts (D=2: 70.8%, D=4: 78.3%) compared to unspecified prompts (D=1: 41.5%). This suggests that baseline models can leverage explicit diversity instructions but struggle with implicit diversity requirements. In contrast, DISCO maintains consistently high UFA performance (97.7-99.7%) across all diversity specifications, effectively eliminating duplicate identities regardless of prompt formulation.

The Global Identity Spread metric reveals a complementary pattern: baseline models generally achieve higher GIS scores on simpler diversity specifications (D=1, D=3) but struggle with complex individual assignments (D=4), where detailed ethnicity specifications appear to constrain their generation diversity. For instance, Flux-Krea drops from 71.9% (D=1) to 52.8% (D=4), and OmniGen2 falls from 48.5% to 29.2%. This indicates that explicit individual constraints paradoxically reduce overall identity diversity in baseline models. DISCO overcomes this limitation, achieving near-perfect GIS scores (98.5-100%) across all prompt types, demonstrating that our compositional reward system successfully handles both implicit and explicit diversity requirements without compromising identity uniqueness.

These patterns confirm that DISCO generalizes robustly across diverse prompt formulations, resolving the fundamental tension between following specific diversity instructions and maintaining overall identity spread that challenges existing models.

Table E.1: Performance across diversity tags (D=1: No tag, D=2: "Diverse faces", D=3: Single ethnicity, D=4: Individual assignments). DisCo shows consistent improvements across all diversity specifications. Green scores indicate the highest results and Red scores indicate the lowest results.

| Model | Count Accuracy | | | | Unique Face Accuracy | | | | Global Identity Spread | | | |
|---|---|---|---|---|---|---|---|---|---|---|---|---|
| | D=1 | D=2 | D=3 | D=4 | D=1 | D=2 | D=3 | D=4 | D=1 | D=2 | D=3 | D=4 |
| **DiverseHumans-TestPrompts** | | | | | | | | | | | | |
| Gemini-Nanobanana | 71.0 | 71.7 | 70.7 | 76.0 | 41.5 | 70.8 | 38.3 | 78.3 | 56.6 | 69.2 | 53.8 | 55.7 |
| Flux-Dev | 70.0 | 70.0 | 69.0 | 74.3 | 47.8 | 41.7 | 47.0 | 56.3 | 64.74 | 58.8 | 67.8 | 62.3 |
| Flux-Krea | 75.0 | 68.0 | 71.3 | 80.3 | 51.3 | 45.3 | 37.5 | 49.2 | 71.9 | 66.66 | 56.9 | 52.8 |
| OmniGen2 | 62.3 | 61.3 | 67.0 | 62.3 | 32.3 | 33.5 | 27.2 | 36.2 | 48.5 | 36.2 | 41.2 | 29.2 |
| DreamO | 71.7 | 70.0 | 70.0 | 70.3 | 31.8 | 20.7 | 27.0 | 45.5 | 52.1 | 40.0 | 51.2 | 43.7 |
| HiDream-Default | 55.7 | 60.0 | 56.0 | 60.0 | 35.7 | 32.0 | 29.3 | 32.5 | 32.4 | 26.2 | 28.3 | 15.9 |
| DisCo | 92.0 | 86.3 | 95.7 | 95.7 | 98.7 | 98.3 | 97.7 | 99.7 | 100.0 | 100.0 | 98.7 | 98.5 |

### E.1.2    GRID SEARCH ON REWARD WEIGHTS

Table E.2 presents results from our systematic exploration of reward weight combinations to understand the sensitivity and optimal balance of our compositional reward function. It is on the Flux-Dev baseline. We apply DisCo finetuning for 300 epochs. The analysis reveals that intra-image diversity ($\alpha$) has the strongest impact on overall performance, with higher weights leading to better Unique Face Accuracy and Global Identity Spread. The group-wise diversity component ($\beta$) shows diminishing returns beyond moderate values, while count accuracy ($\gamma$) requires careful balancing to avoid

over-penalization. Quality component ($\zeta$) demonstrates that moderate values suffice for maintaining perceptual quality without sacrificing diversity objectives. We pick the optimal configuration $\alpha = 0.5$, $\beta = 0.1$, $\gamma = 0.3$, $\zeta = 0.2$ for our final experiment. Note that the final results in Section 4(at 480 epochs) are slightly different, as the results in this Table are all compared at 300 epochs to stay consistent.

Table E.2: Ablation study on reward weight parameters. Results are for DisCo(Flux-Dev). Each row shows the effect of different weight configurations on overall performance metrics. Our selected hyperparameter configuration is represented in the Blue row.

| Reward Weights | | | | Metrics | | | |
|---|---|---|---|---|---|---|---|
| $\alpha$ | $\beta$ | $\gamma$ | $\zeta$ | Count | Unique Face | Global Identity | HPS |
| (Intra-Img) | (Grp-wise) | (Count) | (Quality) | Accuracy | Accuracy (UFA) | Spread (GIS) | |
| 0.3 | 0.1 | 0.2 | 0.4 | 84.2 | 90.1 | 77.7 | **33.8** |
| 0.3 | 0.1 | 0.4 | 0.2 | 81.2 | 86.3 | 87.7 | 33.0 |
| 0.5 | 0.1 | 0.2 | 0.2 | 88.3 | **96.7** | 97.4 | 33.6 |
| 0.5 | 0.2 | 0.3 | 0.0 | **90.0** | 95.3 | **98.1** | 29.3 |
| 0.5 | 0.0 | 0.3 | 0.2 | 87.8 | 94.5 | 80.1 | 33.7 |

### E.1.3 INTRA-IMAGE DIVERSITY AGGREGATION FUNCTION ANALYSIS

Table E.3 compares different aggregation strategies for computing the intra-image diversity reward when multiple faces are detected within a single image. We perform this analysis on the harder-to-converge DisCo-Krea setup. The results are on DiversePrompts. The choice of aggregation function impacts both convergence behavior and final performance characteristics.

Table E.3: Comparison of aggregation functions for intra-image diversity reward computation. Results show performance on Flux-Krea baseline. Blue represents the selected aggregation function.

| Aggregation | Count | Unique Face | Global Identity | HPS |
|---|---|---|---|---|
| Function | Accuracy | Accuracy (UFA) | Spread (GIS) | Score |
| max() | 83.8 | **80.1** | **84.1** | **32.9** |
| mean() | **84.3** | 77.8 | 82.3 | 32.8 |
| min() | 84.1 | 74.2 | 77.7 | 32.9 |

Using max() aggregation drives the network toward eliminating the most similar face pair within each image, penalizing any identity overlaps. This approach, particularly when combined with curriculum learning, enables faster convergence and achieves lesser overlapping identities. It essentially implements a "fix the worst violation" strategy that systematically eliminates duplicate identities.

In contrast, mean() aggregation optimizes for low average similarity across all face pairs, which can result in suboptimal solutions where multiple moderate violations persist rather than being eliminated entirely. It converges more slowly and allows identity overlaps to remain, as the model can satisfy the average similarity constraint without addressing individual duplicate pairs. The min() function shows the poorest performance, as it focuses on the least similar pair and provides insufficient pressure to address problematic duplicates.

### E.2 FINAL RUN REWARD CURVES

Figure E.1 demonstrates the training progression of DISCO across all four reward components throughout the learning process. The curves show consistent improvement in intra-image diversity, group-wise diversity, count accuracy, and HPS quality metrics during both training and evaluation phases. While training rewards continue to grow post 500 epochs, the model generates diminishing returns on the testset post 480 epochs. The total training time for a single run is 13 hours.

### E.2.1 COMPUTATIONAL ANALYSIS

Table E.4 presents a comprehensive comparison of computational efficiency across all evaluated models. We report average performance scores from our multi-human generation benchmarks along-

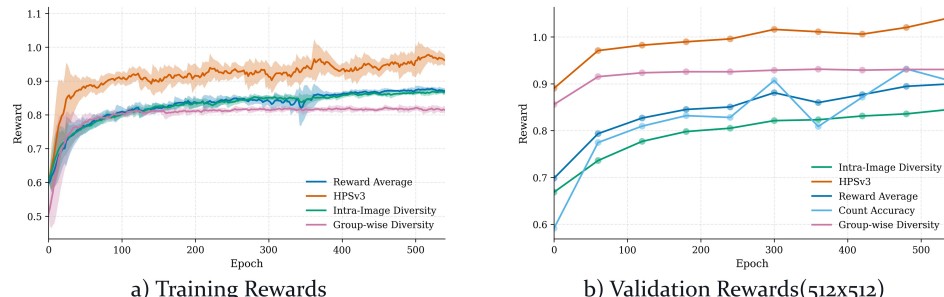

a) Training Rewards        b) Validation Rewards(512x512)

Figure E.1: **DisCo training and evaluation reward curves.** As observed, we notice a steady improvement in all four rewards during training and inference.

side timing measurements to assess the quality-efficiency trade-off. For proprietary models, we report API response times including network latency, while for open-source models we measure local inference runtime on standardized hardware (NVIDIA H100) for generating a 1024×1024 image with default sampling steps.

DISCO demonstrates an excellent balance between generation quality and computational efficiency. While proprietary models like GPT-Image-1 achieve competitive scores, they incur ongoing API costs and lack deployment flexibility. Gemini-Nanobanana offers faster API responses but with significantly lower generation quality. Among open-source alternatives, DISCO variants significantly outperform existing methods in generation quality while maintaining identical inference times to their respective base models. This makes DISCO particularly attractive for applications requiring both high-quality multi-human generation and practical deployment constraints, offering superior performance without sacrificing efficiency.

Table E.4: Computational efficiency comparison across all evaluated models. Average scores are from DiverseHumans-TestPrompts benchmark. Runtimes are measured on NVIDIA H100 for open-source models.

|  | Model | Average Score | API Time (seconds) |
|---|---|---|---|
| Proprietary | Gemini-Nanobanana | 60.0 | 7 |
|  | GPT-Image-1 | 78.7 | 28 |
|  |  | Average Score | Runtime (seconds) |
| Open-Source | HiDream | 46.2 | 22 |
|  | Qwen-Image | 60.1 | 23 |
|  | OmniGen2 | 48.8 | 14 |
|  | Flux | 56.0 | 9 |
|  | Flux-Krea | 57.8 | 6 |
| Ours | DISCO(Flux) | 81.7 | 9 |
|  | DISCO(Krea) | 76.8 | 6 |

## E.3    QUALITATIVE RESULTS

### E.3.1    VISUALIZING GLOBAL IDENTITY SPREAD

Figure E.3 demonstrates the effectiveness of DISCO in achieving global identity diversity compared to the baseline Flux-Dev model. The visualization shows three different prompts, each generating six images using consistent random seeds. The baseline Flux model exhibits significant identity overlap both within individual images and across the generated set, with many faces appearing similar or identical. In contrast, DISCO fine-tuning successfully pushes facial identities apart in the embedding space, resulting in visually distinct individuals across all generations while maintaining high visual quality and prompt adherence.

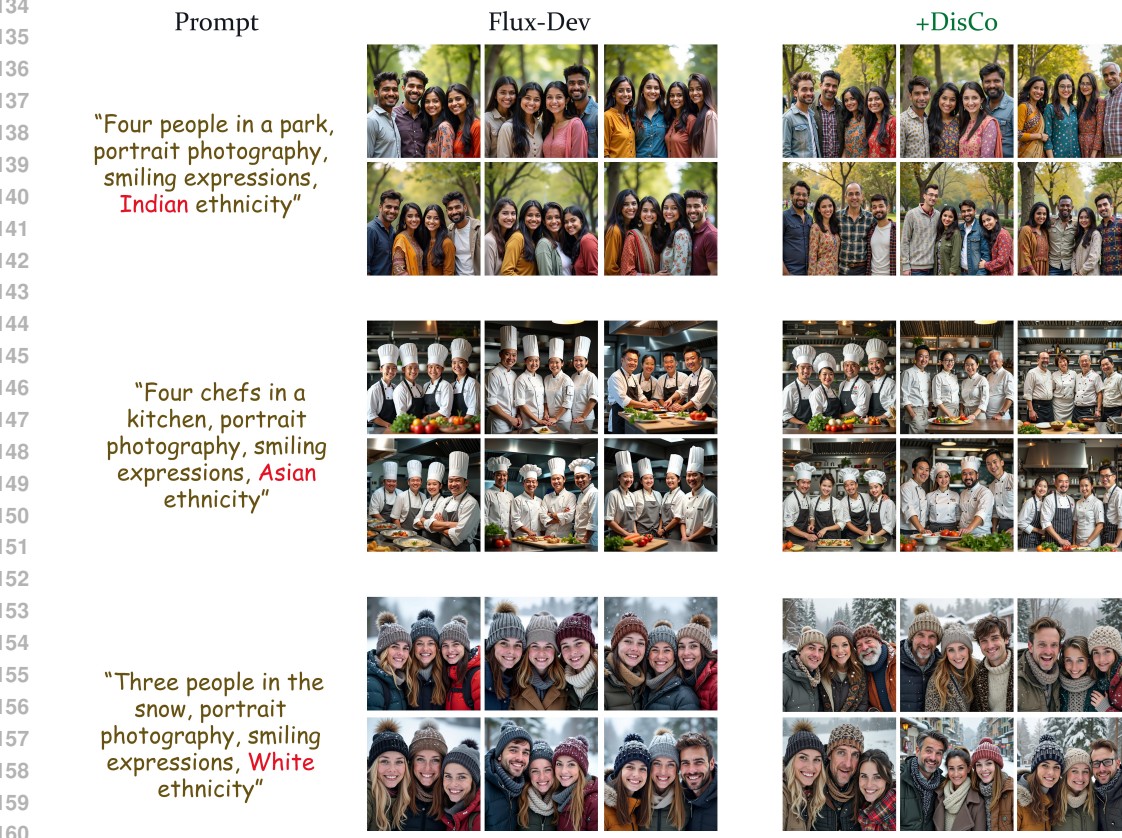

Figure E.2: **DISCO v/s Flux-Dev** As observed in this Figure, we visualize three prompts of people containing the same ethnicity, over six consistent seeds for DisCo and Flux. As observed, Flux results not only generate overlapping identites in the same image, but generate similar looking people across the dataset. However, DisCo finetuning pushes the faces further from each other.

### E.3.2 RESULTS ON FLUX-KREA

Figure E.3 showcases the qualitative improvements achieved by applying DISCO fine-tuning to the Flux-Krea baseline model. The comparison demonstrates that our approach successfully addresses identity consistency issues present in existing methods while preserving the aesthetic qualities that make Flux-Krea distinctive. The generated images show clear improvements in generating distinct individuals without duplicate identities, accurate person counts matching prompt specifications, and maintained perceptual quality. These results validate that our method generalizes effectively across different base models while preserving their unique characteristics.

### E.3.3 VISUAL EFFECTS OF COUNT AND HPS REWARD COMPONENTS

Figure E.4 illustrates the visual effects of our count and HPS reward components in addressing common failure modes during DISCO training. These components are essential for preventing visual artifacts and ensuring realistic multi-person generation.

The top row of Figure E.4 demonstrates the visual improvements achieved through HPS rewards. Without perceptual oversight, models produce unnatural grid-like face arrangements that technically satisfy count and diversity requirements but result in unrealistic images. The progression from no HPS to HPSv2 to HPSv3 shows systematic improvement in visual coherence, with HPSv3 producing the most aesthetically pleasing results and minimal degradation artifacts.

The bottom row illustrates the visual impact of count rewards: as shown in Figure E.4, without count control the model generates fewer people than requested (5 instead of 7) to avoid the challenging

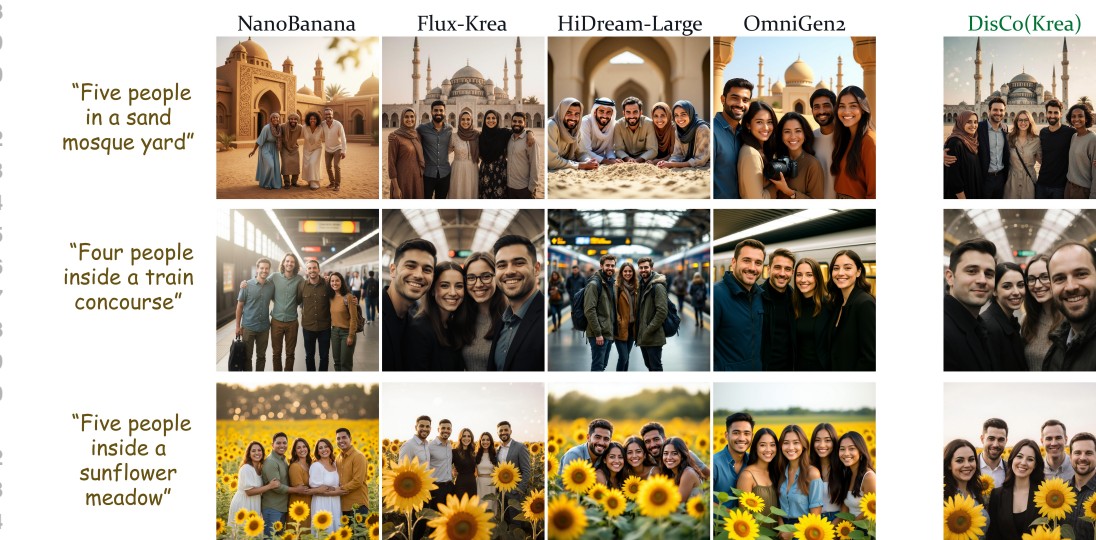

Figure E.3: **DISCO-KREA v/s Related Work** DISCO finetuning applied to Flux-Krea improves performance over current baselines to generate results which consistently generate accurate people without overlapping identity, without a hit in perceptual quality.

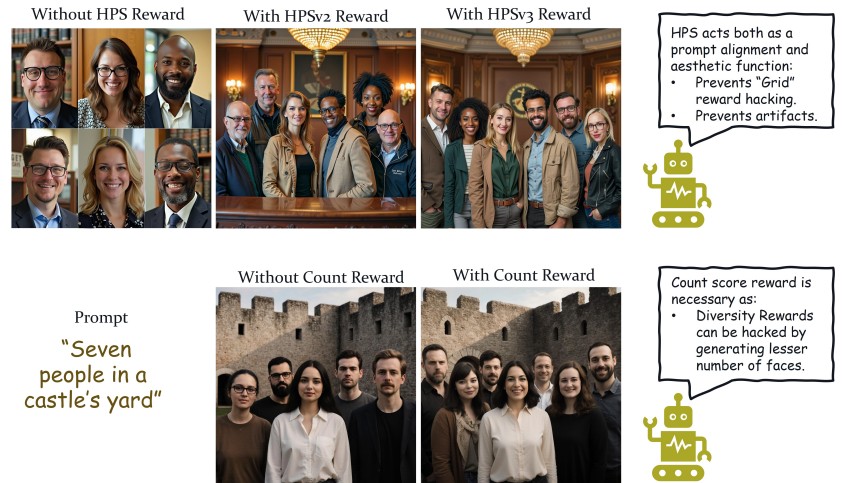

Figure E.4: **Visual effects of count and HPS reward components.** *Top row:* HPS rewards reduce grid artifacts and improve visual quality, with HPSv3 achieving the most natural arrangements. *Bottom row:* Count rewards ensure correct number generation (7 people instead of 5) while maintaining visual coherence.

task of creating multiple distinct identities. Our count reward component directly addresses this by ensuring the correct number of people are generated while maintaining visual quality.

Together, these reward components ensure that our approach produces visually coherent and accurate multi-person generations, preventing both under-generation and visual artifacts that can emerge from optimizing individual objectives in isolation.

# F    LIMITATIONS AND FUTURE WORK

Our approach relies on face detection and face-embedding similarity; as such, failure cases can arise under heavy occlusion, extreme poses, partial profiles, or when faces are very small. Future directions for this line of work include integrating body/appearance cues beyond faces (e.g., re-identification or whole-body embeddings), extending DISCO to videos with spatiotemporal identity consistency, extending disco to other (diverse in nature) concepts such as animals, learning adaptive curricula, and exploring human-in-the-loop or active reward shaping. Finally, we aim to study fairness and demographic balance more explicitly, and to evaluate robustness to higher person counts.