# OpenReview forum: "DisCo: Reinforcement with Diversity Constraints for Multi-Human Generation"
_ICLR.cc/2026/Conference — ICLR 2026 Conference Withdrawn Submission_

### Official Review · Reviewer_zP3i · 2025-10-18

**Soundness:** 2
**Presentation:** 3
**Contribution:** 2
**Rating:** 2
**Confidence:** 4

**Summary:**

This paper introduces DISCO, which utilizes RL to solve the problems like duplicating faces, merging identities, and miscounting individuals in prevailing image generation models. It features a group-wise diversity reward and a single-stage curriculum strategy. This method outperforms prevailing models on a self-built benchmark and an open-sourced benchmark.

**Strengths:**

- The work represents a commendable effort to leverage reinforcement learning for image generation. Tackling controllability through an RL framework is a non-trivial and potentially valuable direction.
- The manuscript is well-written and clearly structured. The exposition is straightforward, and the technical descriptions are easy to understand.
- Empirical results suggest that the proposed method performs adequately for coarse-grained control tasks, such as specifying the number of human figures or preventing the id confusion of faces in generated images.

**Weaknesses:**

- The methodological design appears relatively basic and does not introduce significant conceptual or technical novelty. In particular, the RL formulation follows standard practices without offering new insights into how RL can be better adapted to the nuances of image generation.
- The success of the method stems primarily from the use of multiple external as reward sources, which limits the paper’s contribution.
- As evidenced by both the method description and the test prompts in the appendix, the approach seems to lack fine-grained control over individual attributes (e.g., hairstyle, clothing style, facial expression). This limitation substantially restricts its practical utility in real-world applications where detailed customization is often required.

**Questions:**

- How were the hyperparameters (e.g., $\alpha,\beta,\gamma,t_\textit{curriculum}$) selected? Was a systematic hyperparameter search performed?
- How stable was the GRPO training process in practice? Did the authors observe frequent training instabilities or policy collapse?
- Have the authors considered architectural or algorithmic extensions to enable more precise, attribute-level control?
- What is the computational efficiency of the RL training ? How much additional training cost does this method incur compared to fine-tuning baselines?

---

### Official Review · Reviewer_Cang · 2025-10-30

**Soundness:** 2
**Presentation:** 2
**Contribution:** 2
**Rating:** 6
**Confidence:** 3

**Summary:**

The paper introduces DISCO, a reinforcement learning framework aimed to improve text-to-image models’ ability to generate multiple distinct humans. Optimizing for facial diversity, correct person counts, and visual fidelity—without extra annotations—DISCO improves identity separation and sets a new benchmark for multi-human image generation.

**Strengths:**

- multi-human prompt adherence is a failing point of text2image modelling. The method presented is a first attempt attacking this problem with reinforcement learning. The method is sensible and clearly improves on this issue quantitatively
- experiment section includes ablation study of loss components showing clear contributions from each component

**Weaknesses:**

- penalizing intra-image facial similarity leads to id divergence might be restrictive at cases, e.g. family pictures, or pictures where we want the same person to appear twice
- for count control the reward is agnostic of the degree of error, flat zero reward independent of the difference of generated vs requested
- reproducibilityy is limited, the method is exposed in great detail, however, code and model weights are not provided
- training data contain images with 2-7 people, does this limit model performance for generating a single person? Evals show results for 1-5 and 2-7 people separately which have great degree of overlap
- More qualtilative results need to be added to the main body of the paper (currently 5 generations are presented) to showcase improvements and failures for specific cases, e.g. single person generation
- Limitations should be included in the main paper in my opinion

(Minor minor)
- repeated "crop crop" in l.214

**Questions:**

- From 3.2 "(ii) discourage reusing the same identity across samples of the same prompt" can you explain what benefit this behaviour brings to the trained model?
- why not try matching face similarities to target images instead of enforcing diversity?
- does DISCO deteriorate model performance for single person generation?

---

### Official Review · Reviewer_jrZd · 2025-11-01

**Soundness:** 3
**Presentation:** 3
**Contribution:** 2
**Rating:** 4
**Confidence:** 4

**Summary:**

This work directly applies GRPO to optimize identity diversity in multi-human generation tasks to resolve collapses such as duplicating faces and merging identities etc. The authors construct a compositional reward function to shape various aspects of multi-human generation and achieves SOTA performance. DisCo appears to be a simple but effective adaptation of RL (GRPO) to multi-human/identity image generation.

**Strengths:**

1. The effectiveness of the compositional rewards is promising, making DisCo outperform GPT. There are many reward terms, but from the reward curves in the Appendix they seem to harmonically improve the model together. Also, the rewards are "annotation-free", which is practical and makes the method scalable.

2. The ablation studies with each reward component and the curriculum is meaningful. It shows that using curriculum for stability is indeed very helpful.

**Weaknesses:**

1. The improvement is valuable but the novelty is incremental: the use of RL to enhance image generation [1], as well as a more specified multi-subject/identity/human generation [2] has been proposed before, and similar diversity rewards are designed in previous work. As the authors also discusses and leverages [2], the novelty of this work to me is the reward engineering, which brings predictable improvement. Concurrent work also adopts similar approach for more complex problems like video generation [3].

2. Then, it would add more value if the authors can provide further insights on analyzing RL algorithms for multi-subject generation, e.g., why GRPO, and comparing between a few PPO-variant RL methods to identify properties/designs that benefit similar tasks, enlightening future work.

[1] Black, Kevin, et al. "Training diffusion models with reinforcement learning." arXiv preprint arXiv:2305.13301 (2023).

[2] Liu, Jie, et al. "Flow-grpo: Training flow matching models via online rl." arXiv preprint arXiv:2505.05470 (2025).

[3] Wu, Tao, et al. "MultiCrafter: High-Fidelity Multi-Subject Generation via Spatially Disentangled Attention and Identity-Aware Reinforcement Learning." arXiv preprint arXiv:2509.21953 (2025).

**Questions:**

Please see more details in Weaknesses. To me, the main concern is limited novelty, the reward engineering works well, the results on multi-human generation are great, but it has been proven that RL with fine-grained rewards can improve image generation tasks. There also lacks more in-depth analysis of RL in such tasks which could potentially draw new insights and observations, making this work incremental. Thus, I cannot recommend acceptance at this point.

---

### Note · Authors · 2025-11-14

**Comment:**

We appreciate the time and efforts the reviewers and AC invested in evaluating our submission. However, after careful consideration, we have decided to withdraw our paper from ICLR 2026.


While we appreciate the feedback provided, we believe that certain key aspects of our work may not have been fully recognized during the review process. For instance, Table E.2 in the supplementary material already includes results from a hyperparameter search, which was specifically requested in the reviews. This suggests that parts of our submission have not been examined by the reviewers.


More importantly, we would like to reiterate the novelty of our contribution. Apart from the intra-image diversity reward, this is the first work to incorporate a reward based on "group-level statistics" for text-to-image models; it explicitly optimizes diversity across the whole generated distribution(this specifically enables ~99% unique faces across the ~5000 requested). Our compositional reward design goes beyond prior efforts by jointly addressing intra-image similarity, cross-sample identity repetition, count accuracy and text/perceptual alignment. Hence, it does not sacrifice perceptual quality, text control or requires additional annotations.


Furthermore, our work is the first ever method to formalize and address a fundamental limitation present in all current text-to-image (T2I) models(including proprietary ones). This is the inability to maintain identity separation across multiple humans, both in the same image and across generated samples. While some reviewers perceived our approach as incremental, we respectfully disagree. One review referenced MultiCrafter, a concurrent ICLR 2026 submission that was not even publicly available before the submission deadline. Unlike reference-image-based generation frameworks (or “editing” models), our method focuses on improving base text-to-image generation without relying on reference images. We show that these long-standing issues can be resolved without compromising existing capabilities for human-level control. Hence, this approach is intended to act as a strong foundation for future work on reference-based generation. While we have upcoming submissions exploring that direction, it falls outside the scope of this paper.


We hope to refine and resubmit this research to a future venue.

**Withdrawal Confirmation:**

I have read and agree with the venue's withdrawal policy on behalf of myself and my co-authors.